# Magnesium Oxychloride Cement Composites Lightened with Granulated Scrap Tires and Expanded Glass

**DOI:** 10.3390/ma13214828

**Published:** 2020-10-28

**Authors:** Milena Pavlíková, Adam Pivák, Martina Záleská, Ondřej Jankovský, Pavel Reiterman, Zbyšek Pavlík

**Affiliations:** 1Department of Materials Engineering and Chemistry, Faculty of Civil Engineering, Czech Technical University in Prague, Thákurova 7, 166 29 Prague 6, Czech Republic; milena.pavlikova@fsv.cvut.cz (M.P.); adam.pivak@fsv.cvut.cz (A.P.); martina.zaleska@fsv.cvut.cz (M.Z.); 2Department of Inorganic Chemistry, Faculty of Chemical Technology, University of Chemistry and Technology, Technická 5, 166 28 Prague 6, Czech Republic; ondrej.jankovsky@vscht.cz; 3Experimental Centre, Faculty of Civil Engineering, Czech Technical University in Prague, Thákurova 7, 166 29 Prague 6, Czech Republic; pavel.reiterman@fsv.cvut.cz

**Keywords:** magnesium oxychloride cement, scrap tires, granulated expanded glass, macrostructural parameters, mechanical resistance, thermal performance, water resistance

## Abstract

In this paper, light burned magnesia dispersed in the magnesium chloride solution was used for the manufacturing of magnesium oxychloride cement-based composites which were lightened by granulated scrap tires and expanded glass. In a reference composite, silica sand was used only as filler. In the lightened materials, granulated shredded tires were used as 100%, 90%, 80%, and 70% silica sand volumetric replacement. The rest was compensated by the addition of expanded glass granules. The filling materials were characterized by particle size distribution, specific density, dry powder density, and thermal properties that were analyzed for both loose and compacted aggregates. For the hardened air-cured samples, macrostructural parameters, mechanical properties, and hygric and thermal parameters were investigated. Specific attention was paid to the penetration of water and water-damage, which were considered as crucial durability parameters. Therefore, the compressive strength of samples retained after immersion for 24 h in water was tested and the water resistance coefficient was assessed. The use of processed waste rubber and expanded glass granulate enabled the development of lightweight materials with sufficient mechanical strength and stiffness, low permeability for water, enhanced thermal insulation properties, and durability in contact with water. These properties make the produced composites an interesting alternative to Portland cement-based materials. Moreover, the use of low-carbon binder and waste tires can be considered as an eco-efficient added value of these products which could improve the environmental impact of the construction industry.

## 1. Introduction

The management, treatment, and disposal of solid waste belong among the main environmental concerns around the globe. With respect to sustainability, the best solutions of waste management are those that minimize the environmental impact at an affordable cost [1,2]. Today, the increasing population leads to an increasing number of vehicles and massive tire production. Consequently, this generates a significant increment in solid waste and, thus, in the volume of scrap tires. Therefore, the sustainable management of end-of-life tires (ELTs) is of the particular importance. According to the ETRMA (European Tyre and Rubber Manufactures’ Association), the tire production within EU countries remained over 4 billion tons throughout the period 2010–2018 and reached a maximum of 5.1 billion tons in 2017 and 2018 [3]. This corresponds to approx. 24% of the total world production. It is estimated that EU countries, USA, Japan, India, and China produce almost 88% of the total number of withdrawn tires around the globe [4].

In Europe, most of the ELTs are recovered as reported in statistics by ETRMA [3]. In 2010, 96% of ELTs were recovered in energy (38%), materials recycling and reuse (40%), reconstruction (8%), and export (10%) [5]. On the other hand, the remaining 0.2 billion tones represents an open field for other commercial reuse of materials based on scrap tires. Moreover, in less-developed countries, the recovery of ELTs is much lower [6]. This opens a broad field for an alternative recycling of scrap tires as their stockpiling has become unacceptable because of the depletion of available sites for landfilling and fire hazards. Basically, ELTs can be managed in a number of forms, such as whole tire, slit tire, shredded or chopped tire, powdered tire, ground rubber, or as a crumb rubber product. In this respect, intensive research has been conducted to find novel and effective methods of ELT treatment. Except the use of scrap tires for energy recovery (tire-derived fuel) and alternative fuel for cement clinker production, where embodied steel sufficiently endows raw flour by iron oxides [7], they have found application in the production of building materials, such as asphalt pavements [8,9,10,11], rubberized asphalt [12,13], rubber-filled concrete [14,15,16,17,18,19], crumb rubber concrete hollow blocks [20], thermal and sound insulation precast concrete panels [21], and rendering mortars with increased durability and thermal insulation performance [22]. As tire-recycling products have low gravity, low thermal conductivity, and are waterproof and durable against harmful environmental action, they have found use in the construction of tunnels, underground passages, highway embankments, acoustic sheets, acoustic flooring foils, etc. [4]. The rubber from worn tires were also found to improve the resistance of cement mortars against water and chloride penetration [23,24]. Recently, rubber particles were found to have potential to be used as bacteria carriers in self-healing concrete [25].

Typically, the high dosage of rubber in concrete led to an excessive drop in mechanical resistance due to the failure of the bond between rubber particles and the silicate matrix, and a reduction in bulk density and elastic modulus. However, processed tires can find use in other types of composites and construction products, where high mechanical strength is not the main technical parameter. For example, they can be applied in the form of lightweight fillers or admixtures providing low permeability for water and dissolved salt, thus ensuring durability in terms of water resistance, ability to absorb deformation without cracking, improved freeze/thaw resistance [26], etc.

As the use of ELTs in Portland cement- and asphalt-based materials was intensively explored within last three decades, we aimed in the presented work at the use of granulated waste tire rubber as lightweight water resistant aggregate in composition of composites made of magnesium oxychloride cement (MOC) as an only binder. The impact of the use of rubber granulate on the technical and functional properties and performance of the researched materials was studied and the findings are discussed and summarized. The research was motivated by the fact the Portland cement industry has an adverse environmental impact which is associated with enormous CO_2_ emissions and natural resources depletion. MOC has many superior properties to those associated with ordinary Portland cement, such as fast hardening, early and high strength, low alkalinity, low density of precipitated products, fire resistance, high bonding ability, CO_2_-neutrality, etc. [27,28,29]. On the other hand, a critical defect of MOC is its limited water resistance which is the cause of its little industrial application [30,31]. As the rubber produced from scrap tires repels water (hydrophobic nature), it is assumed it will restrict water transport in the MOC matrix and, thus, ensure its resistance with regard to water damage. It is also expected that the lightweight rubber will help to enhance thermal insulation performance of the final composites. Similar benefits should also be provided by granulated expanded glass, which was used in composite mixtures as a partial substitute of processed waste tires. The motivation of the paper can be summarized as: the design, development, and testing of lightweight eco-efficient MOC composites with improved water resistance and thermal insulation capability. The novelty of the presented study is evident as the use of recycled waste tires in MOC composites was only rarely studied up to now, and data on hygric and thermo-physical properties of MOC/granulated rubber composites are not available at all.

## 2. Materials and Methods

For preparation of MOC binder, MgCl_2_·6H_2_O (purity > 99%, Lach-Ner, s.r.o., Neratovice, Czech Republic) and MgO produced by Styromagnesit Steirische Magnesitindustrie Ltd. (Oberdorf, Austria) were used. The chemical composition of caustic MgO was following: MgO_2_ (81.5 wt%), SiO_2_ (3.6 wt%), Al_2_O_3_ (5.5 wt%), CaO (4.7 wt%), Fe_2_O_3_ (4.0 wt%), SO_3_ (0.6 wt%).

In the reference composite (MOC-S) silica sand, a particle size in the range of 0–2 mm was dosed as the only filler. It was delivered by Filtrační písky, spol. s.r.o. (Chlum u Doks, Czech Republic). Silica sand was used as a mixture of three sand fractions (0.063–0.5 mm, 0.5–1.0 mm, 1.0–2.0 mm) which were mixed in a mass ratio 1:1:1. The main constituents of sand were SiO_2_ (97.1 wt%), Al_2_O_3_ (2.3 wt%), MgO (0.3 wt%), and TiO_2_ (0.2 wt%). The chemical composition of silica sand and MgO was measured using an ED-XRF spectrometer ARL QUANT’X (Thermo Fisher Scientific, Waltham, MA, USA). For data collection and analysis, UniQuant 5 software (Thermo Scientific, Milan, Italy) was applied.

The rubber granulate was produced by Montstav CZ s.r.o. (Dolní Rychnov, Czech Republic) by mechanical processing of waste tires, which is the most frequently used technology for rubber reclaiming [32]. The recycling of ELTs starts with the pulling out of the heel rope, which would cause an excessive wear of the crusher cutting tool. Thereafter, the tire is transported to the knife mill hopper by using a conveyor belt. The mill grinds the tire to so-called “chips” of different dimensions, which are subsequently sorted out in accordance with the actual requirements; the largest pieces returning to the mill. The process of secondary mechanical crushing is carried out in a different device which has the specified design of the crushing knifes. Originating grit continues through a magnetic sorter to omit steel fibers. Resulting rubber grit is sorted to the various grades. In our case, the maximum size of rubber granulate was 2 mm (mix of fractions 0–1 mm, 1–2 mm). The processing of waste tires by Montstav CZ s.r.o. is shown in Figure 1.

Expanded glass granulate, commercial name Liaver (Liaver GmbH and Co. KG, Ilmenau, Germany) is produced from recycled glass and sintered at temperatures from 750 °C to 900 °C in a rotary kiln. It is composed of SiO_2_ (71 ± 2 wt%), Al_2_O_3_ (2 ± 0.3 wt%), Na_2_O (13 ± 1 wt%), Fe_2_O_3_ (0.5 ± 0.2 wt%), CaO (8 ± 2 wt%), MgO (2 ± 1 wt%), K_2_O (1 ± 0.2 wt%). The data was taken from the technical sheet of the product). In our case, we mixed four Liaver fractions originally manufactured to obtain similar particle size distribution as that of silica sand. The wt% of particular Liaver fractions were as follows: 0.1–0.3 mm 23%, 0.25–0.5 mm 22%, 0.5–1 mm 25%, 1–2 mm 30%. Both alternative lightweight fillers, i.e., shredded rubber and expanded glass granulate, are shown in Figure 2. In the case of granulated rubber, particles and plates of different size, thickness, and shape were observed. The particles of glass granulate were oval- and sphere-shaped, and their size corresponded with the measured particle size curve.

Silica sand, granulated rubber, and Liaver were characterized by specific density *ρ*_sf_ (kg∙m^−3^) that was measured using a Pycnomatic ATC helium pycnometer (Porotec, Hofheim, Germany). For applied fillers, powder density *ρ*_p_ (kg∙m^−3^) and thermal parameters were measured in loose and compacted states. The compaction was conducted with a vibration exciter for 90 s. The powder density was determined by the use of graduated cylinder (sample volume), and precise laboratory scales (sample mass) A&D GX 20,021 (A&D weighing, Adelaide, Australia). The measurement of fillers thermal conductivity *λ*_f_ (W∙m^−1^∙K^−1^) and volumetric heat capacity *C*_vf_ (J∙m^−3^∙K^−1^) was performed with an Isomet 2114 apparatus (Applied Precision, Bratislava, Slovakia) equipped with a needle probe for measurement of granular and powdered substances. The apparatus operates on a nonstationary heat impulse technique [33] with the accuracy 5% of reading + 0.001 W∙m^−1^∙K^−1^ for *λ* in the range of 0.015–0.70 W∙m^−1^∙K^−1^, and 10% of the reading for *λ* ranging from 0.70 W∙m^−1^∙K^−1^ to 6.0 W∙m^−1^∙K^−1^. The volumetric heat capacity was obtained with accuracy 15% of reading +1 × 10^3^ J∙m^−3^∙K^−1^. The grain size analysis was also conducted by standard sieve method as prescribed in EN 933-1 [34]. The used sieves had mesh dimension 0.063, 0.125, 0.25, 0.5, 1.0, and 2.0 mm, respectively.

Rubber granulate was used as a full (material labelled MOC-WT) and partial silica sand replacement. The replacement ratio was 90%, 80%, and 70% by volume. In composites, where silica sand was only partially substituted, the rest of filler volume was supplemented with Liaver. These composites were marked MOC-WT+L10, MOC-WT+L20, and MOC-WT+L30, respectively. The composition of the investigated MOC-based composites is introduced in Table 1. These were mixed and prepared in accordance with the EN 14016-2 [35] that defines dosage of MgO, MgCl_2_, and water for preparation of pastes based on MgO of p.a. purity. Therefore, the dosage of MgO in composite mixtures was higher in order to ensure the precipitation of MOC phases. We used a similar approach in our recently published paper [36], where the design and testing of MOC composites with coal fly ash admixture was presented.

The water/binder ratio was 0.26 and was kept similar for all composite mixtures.

The fresh composite mixtures were casted into plastic molds that were oiled with mineral oil. The samples were prisms having dimensions 40 mm × 40 mm × 160 mm. After 24 h the samples were unmolded and left to freely cure for 13 days in laboratory at *T* = (23 ± 2) °C, *RH* = (50 ± 5) %.

The testing methods were chosen in order to get complex information on the impact of the use of rubber and expanded glass granulates on technical, functional, and performance parameters of the developed composites and to characterize their resistance against harmful water action.

For fresh composite mixtures, workability was tested using a flow table test.

The hardened composites were characterized by the measurement of their macrostructural parameters, mechanical parameters, hygric, and thermal properties. The durability of the examined materials with respect to possible moisture damage was also the subject of the research. This broad experimental campaign was designed and conducted in such a way to get detailed knowledge of the properties of the developed composites which should help to assess their application potential. Bulk density, specific density, and total open porosity were the investigated macrostructural parameters. Before the particular tests, samples were dried at 30 °C in a vacuum dryer Vacucell (BMT, Brno, Czech Republic). The dry bulk density *ρ_b_* (kg·m^−3^) was tested in accordance with the EN 1015-10 [37]. The expanded combined uncertainty of the bulk density assessment was 1.4%. The specific density *ρ_s_* (kg·m^−3^) was measured on a helium pycnometry principle similarly as in testing of fillers (see above). The expanded combined uncertainty of this test was 1.2%. For the measurement, the fragments of samples used for strength tests were used. The typical sample mass was 2.5–3.5 g. The total open porosity *Ψ* (%) was calculated as:(1)$$\Psi =\left(1-\frac{\rho_{b}}{\rho_{s}}\right)\cdot 100$$
with the expanded combined uncertainty 2.0%.

Flexural strength *f*_f_ (MPa) was determined in a three point bending test arrangement on original casted prisms. The compressive strength *f*_c_ (MPa) was measured on the specimen fragments from flexural strength testing. The loading area was 40 mm × 40 mm. The strength tests were realized in accordance with the EN 1015-11 [38]. The expanded combined uncertainty of both strength tests was 1.4%. The stiffness of researched composites was characterized by dynamic elastic modulus *E*_d_ (GPa). It was measured using an ultrasonic data logger Vikasonic (Schleibinger Geräte, Buchbach, Germany) with the uncertainty of 2.3%. During the test, the dry casted prisms were placed between ultrasonic transducers (transmitter and receiver, 54 kHz) and the velocity of ultrasonic pulse *ν* (m/s) travelled in the sample was obtained based on sample length *l_s_* (m) and transmission time *t_t_* (s). The experimental setup of the ultrasonic test is apparent from Figure 3. Based on the measured ultrasonic pulse velocity *ν*, the dynamic elastic modulus was determined as defined in Equation (2) [39]:(2)$$E_{d}=\rho_{b}{\left(\frac{l_{s}}{t_{t}}\right)}^{2}=\rho_{b}\nu^{2}.$$

As MOC is an aerial binder, MOC-based products usually deteriorate in contact with water due to the decomposition of precipitated products of binder hardening [40,41,42]. Therefore, the hygric properties of the developed materials were of particular importance in the presented experimental campaign. As basic hygric performance characteristic describing the water imbibition in porous media, the 24 h water absorption *W*_a_ (%) was measured following the procedure specified in the EN 13755 [43]. The expanded combined uncertainty of the assessment of parameter *W*_a_ was 1.2%. In the capillary absorption test, water absorption coefficient *A*_w_ (kg∙m^−2^∙s^−1/2^) was measured [44]. The samples had dimensions of 40 mm × 40 mm × 70 mm, and their lateral sides were coated with epoxy resin in order to ensure 1-D water transport. The water in reservoir stayed 3–5 mm above the sample bottom as recommended by Feng et al. [45]. The evaluation of measured data and the experimental arrangement followed the EN 10115-18 [46]. The water absorption coefficient was calculated using a one-tangent method [47], and the capillary moisture content *w*_cap_ (kg∙m^−3^) was derived as the maximum of water imbibition curve obtained in a capillary absorption test. Usually, *A*_w_ is measured at (21 ± 2) °C. In our case, the laboratory temperature reached (25 ± 2) °C. Therefore, the temperature compensation of assessed *A*_w_ (25 °C) value was done as introduced in Equation (3):(3)$$A_{w}\left(21\;{}^{\circ}C\right)=\frac{A_{w}\left(T\right)}{0.0112\left(T-273.15\right)\;+\;0.7756}$$
where *T* (K) is the absolute temperature [45].

Based on the values of the water absorption coefficient *A*_w_ (21 °C) and capillary moisture content *w*_cap_, the apparent moisture diffusivity *D*_app_ (m^2^∙s^−1^) was calculated using Equation (4) as originally proposed by Kumaran [48]:(4)$$D_{\mathrm{app}}={\left(\frac{A_{w}\;\left(21\;{}^{\circ}C\right)}{w_{\mathrm{cap}}}\right)}^{2}$$

The expanded combined uncertainty of the capillary absorption test was 2.3% for *A*_w_ (21 °C), 1.8% for *w*_cap_, and that of the apparent moisture diffusivity assessment was 3.5%.

Heat transport and storage were described with thermal conductivity *λ* (W∙m^−1^∙K^−1^), thermal diffusivity *a* (m^2^∙s^−1^), and volumetric heat capacity *C*_v_ (J∙m^−3^∙K^−1^) measurements, which were carried out by a transient plane source technique [49,50]. A TPS 1500 hot-disk thermal constant analyzer (Hot Disk AB, Göteborg, Sweden) with a Kapton-insulated sensor was used. Tests were conducted at room temperature (23 ± 2) °C. The dried specimens had dimension 40 mm × 40 mm × 70 mm. The sensor was placed between two parallel specimens, the cross-section of the specimens contact was 40 mm × 40 mm (see Figure 4 for the measurement arrangement).

The durability of the developed lightweight composites in terms of water damage was evaluated using the water resistance coefficient [51] that was calculated using Equation (5):(5)$$\alpha_{w}=\frac{f_{c24w}}{f_{c}}$$
where *f_c24w_* is the compressive strength of composites immersed in water for 24 h.

Optical microscopy of composite samples and lightweight aggregates was performed by a Navitar macro-optics (Rochester, NY, USA) microscope with optical zoom up to 110× and recorded with a Sony 2/3” digital camera having a resolution of 5 Mpixels. The sample was illuminated by a white LED ring light source with individually addressable segments and intensity. NIS-Elements BR 5.21.02 software (Prague, Czech Republic) with an extended depth of focus module (EDF) was used for imaging and analysis of the samples.

## 3. Results and Discussion

The specific density of used fillers *ρ*_sf_ is presented in Table 2. The highest density had silica sand which was in agreement with its dense and solid character. Both rubber and glass granulates had low specific densities due to their high porosity and the nature of their origin. For these materials, the effect of particle size was quite visible. Typically, the coarser granules yielded lower specific density. This feature was more distinct for Liaver particles rather than for the processed rubber.

There are many types of lightweight aggregates with density ranging from 50 kg∙m^−3^ (expanded perlite) to 1000 kg∙m^−3^ (clinker) or even more [52]. These enable the production of construction composites with a wide range of properties meeting specific technical and functional requirements. The dry powder density of loose fillers and those compacted for 90 s is given in Table 3. For these materials, thermal parameters are also introduced.

The particle size distribution of silica sand, waste tire rubber, and Liaver is graphed in Figure 5. The mixture of Liaver granules gave similar grain size distribution to silica sand. Particles of rubber granulate were slightly coarser compared to two other tested aggregates, but their maximum size was identical.

Macrostructural parameters of hardened composites are summarized in Table 4. The reference material MOC-S exhibited high dry bulk density, specific density and, thus, low porosity, which are typical for MOC-based materials [53]. The replacement of silica sand with processed rubber greatly decreased both the specific density and bulk density of rubberized composites which resulted in the increased porosity of these materials. Due to the lack of standards for MOC-based materials, we used standard EN 206-1 [54] for the classification of the developed materials, which covers structural lightweight concrete. Lightweight concrete must have an oven-dry density in the range 800–2000 kg∙m^−3^, which is then divided into density classes with a span of 200 kg∙m^−3^ [55]. In this respect, composites MOC-WT and MOC-WT+L10 were categorized into class LC 1.4, and materials MOC-WT+20, MOC-WT+30 into class LC 1.3, respectively.

In Table 4, the workability of fresh mixtures is characterized by the measured spread diameter. The flow diameter greatly dropped with the use of rubber granulate, which was assigned to its uniform particles size, shape, and their rugged surface. On the other hand, the use of Liaver improved slightly the workability of fresh mixtures compared to full substitution of silica sand with processed waste tires. This was due to oval-shaped and smooth granules of expanded glass.

The parameters that characterize the strength and stiffness of examined hardened composites are given in Table 5. The control material made of silica sand and MOC binder showed high mechanical strength and dynamic elastic modulus, which was in agreement with previously published papers [56,57]. One must take into account the high mechanical resistance which was obtained after 14 days of curing, and high flexural/compressive strength ratio that are superior properties of MOC binder to ordinary Portland cement [58].

The sand substitution by lightweight aggregate resulted in a significant decrease in mechanical resistance. It was due to the lower powder density and specific density of both rubber and expanded glass granulates in comparison with silica sand. This finding was also in agreement with the porosity data, as the higher porosity brought lower mechanical strength and stiffness. As the density of Liaver was lower than that of shredded rubber, the compressive strength of composite MOC-WT was slightly higher compared to materials with Liaver addition. On the other hand, composites with expanded glass yielded higher modulus of elasticity, but the differences were typically small, similarly as observed for the flexural strength values. The minimum strength class of lightweight concrete given in the EN 206-1 [54] is LC8/9 that refers to a characteristic cylindrical compressive strength of 8 MPa and cubic strength of 9 MPa. This condition met all lightweight composites that nearly satisfied criteria of the strength class LC12/13 which is required by the design standard EN 1992-1-1 [59] for the structural lightweight concrete.

The parameters characterizing the water imbibition and storage in researched composites are shown in Table 6. These parameters are of the particular importance as MOC-based materials are generally considered as susceptible to moisture attack and damage [60]. The 24 h water absorption and both water transport parameters were very low, which pointed only to limited moisture ingress. In case of MOC-S, the rate of water imbibition was restricted by the low open porosity of this material. The limited moisture transport in lightweight composites was due to the non-absorbing and repellent character of rubber and glass granulates which surpassed the effect of higher porosity of these materials compared to that of the reference composite.

The thermophysical properties of the investigated composites determined on dry samples using a hot disk apparatus are introduced in Table 7. Due to the low porosity of MOC-S its thermophysical parameters were high. The ability to transport and store heat was similar as that presented by Záleská et al. [61]. Here, it must be stated the thermal behavior of MOC-based materials was only rarely studied up to now, although numerous studies reported on their mechanical properties and precipitation of hydrated-like products within an MOC setting. The lightened composites exhibited decelerated heat transport and dropped heat storage, whereas the measured thermal parameters were results of the competition of two effects, namely of high porosity and thermal properties of lightweight fillers themselves. The use of Liaver slightly decreased the *λ* and *C***_v_** values in comparison with the composites with full replacement of silica sand by rubber granulate. This corresponded with the thermal parameters of both aggregates, and higher porosity of Liaver enriched materials.

Figure 6 shows the control compressive strength and the compressive strength of composites placed for 24 h in water. The water resistance coefficient is introduced in Table 8. When being immersed for 24 h in water, the compressive strength of all investigated samples increased. This means the water immersion acted as an accelerating curing method. On a similar performance of MOC-based materials reported, Yu et al. [62] observed an increase in the mechanical resistance up to seven days of water exposure. On the other hand, authors observed that the compressive strength decreased with prolonged water immersion due to the decomposition of the main precipitated products [63,64].

The photographs of hardened samples provided by optical microscopy are presented in Figure 7. The dense structure of reference composite MOC-S is particularly visible. In lightweight composites, the incorporation and distribution of rubber and Liaver granules in the MOC matrix was observed.

## 4. Conclusions

Novel lightweight composites for construction use were developed and tested. The materials were made of MOC and processed waste tires which were partially substituted with granulate of expanded glass. The use of a lightweight filler produced from waste tires can help to reduce the continuously-increasing amount of this type of waste and, thus, increase the sustainability of the construction industry. Moreover, as MOC is considered to be a low-carbon binder, the manufactured composites represent an eco-efficient alternative to Portland cement-based materials whose production enormously depletes natural resources and contributes substantially to the generation and emission of greenhouse gases. The use of processed waste also brings financial benefits. Based on the obtained experimental data, the following findings were highlighted:(i)application of processed ELT and glass granulate resulted in an increased porosity and great drop in bulk density;(ii)the develop lightweight materials had sufficient mechanical strength and stiffness which were suitable also for structural applications;(iii)the use of non-absorbing rubber particles and glass granulate repelled the water ingress, the composites, therefore, exhibited low permeability for water;(iv)the both lightweight aggregates used improved the thermal insulation performance of the manufactured composites compared to reference material with silica sand as the only filler;(v)because of the low water penetration, the novel materials were durable with respect to moisture damage.

Summarizing the results of the conducted tests and analyses, the rubberized MOC-based composites may represent a basis for modern construction materials with improved durability and thermal insulation performance. These composites thus represent an interesting alternative to traditional building products.

## Figures and Tables

**Figure 1 materials-13-04828-f001:**
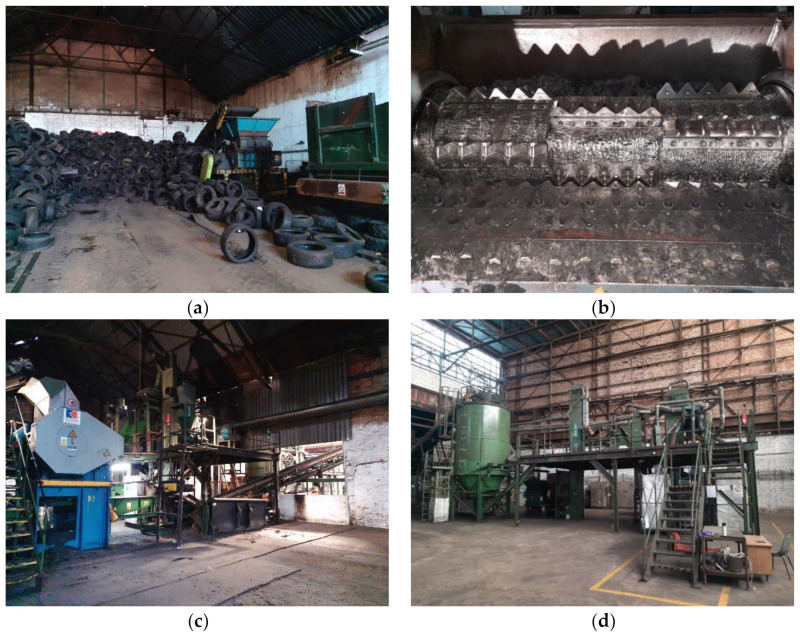
The mechanical processing of waste tires, (**a**) tires’ stocking and transportation, (**b**) knife mill, (**c**) sorting of rubber chips, (**d**) secondary crushing.

**Figure 2 materials-13-04828-f002:**
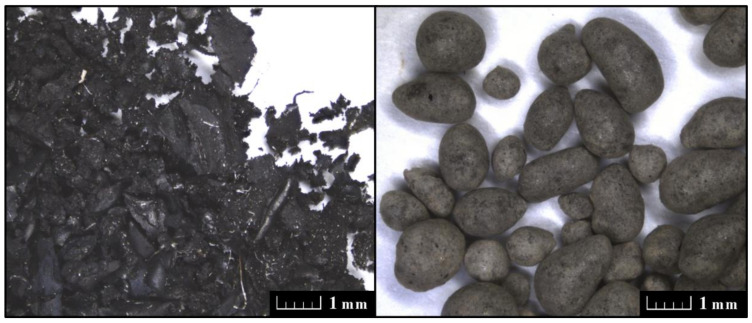
Rubber granulate (**left**) and expanded glass granulate (**right**), scale bar: 1 mm.

**Figure 3 materials-13-04828-f003:**
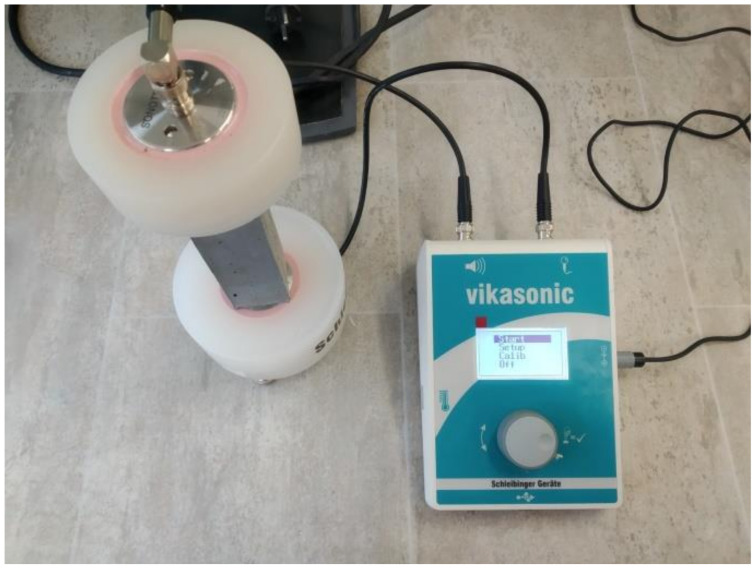
Measurement of the dynamic elastic modulus.

**Figure 4 materials-13-04828-f004:**
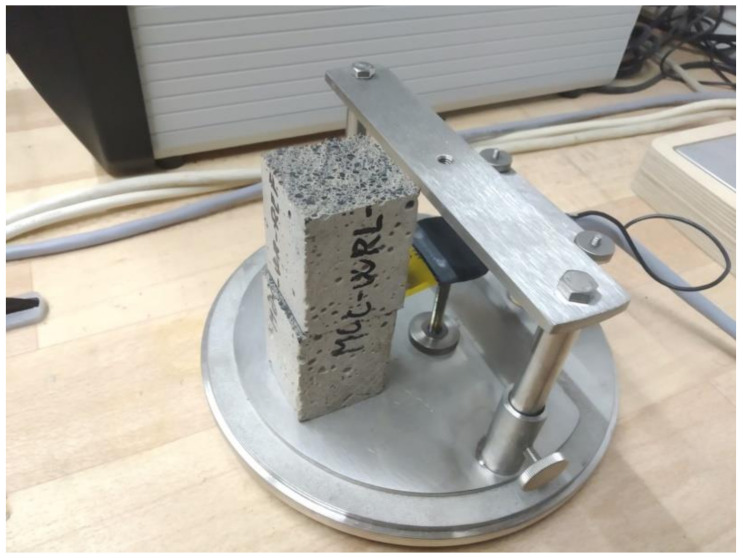
Experimental setup for thermophysical parameter measurement.

**Figure 5 materials-13-04828-f005:**
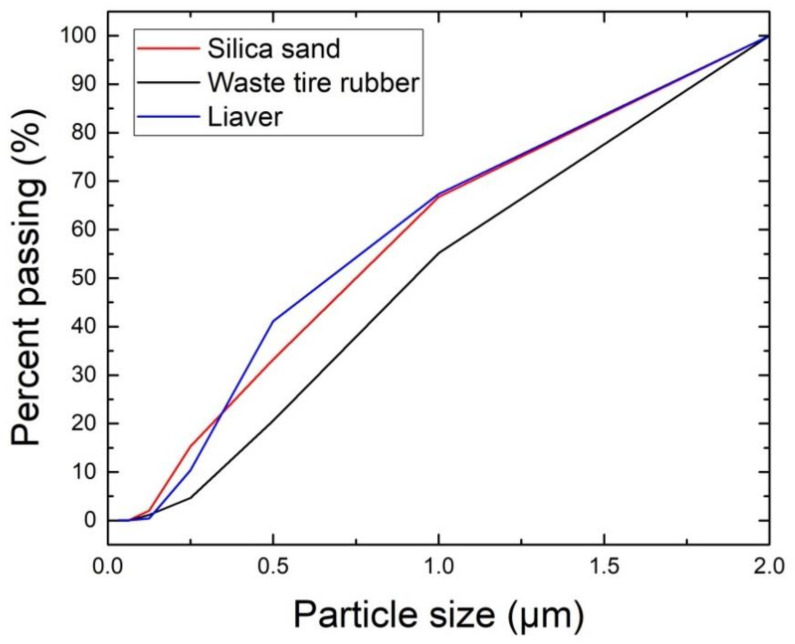
Particle size distribution of silica sand, waste tire rubber, and Liaver.

**Figure 6 materials-13-04828-f006:**
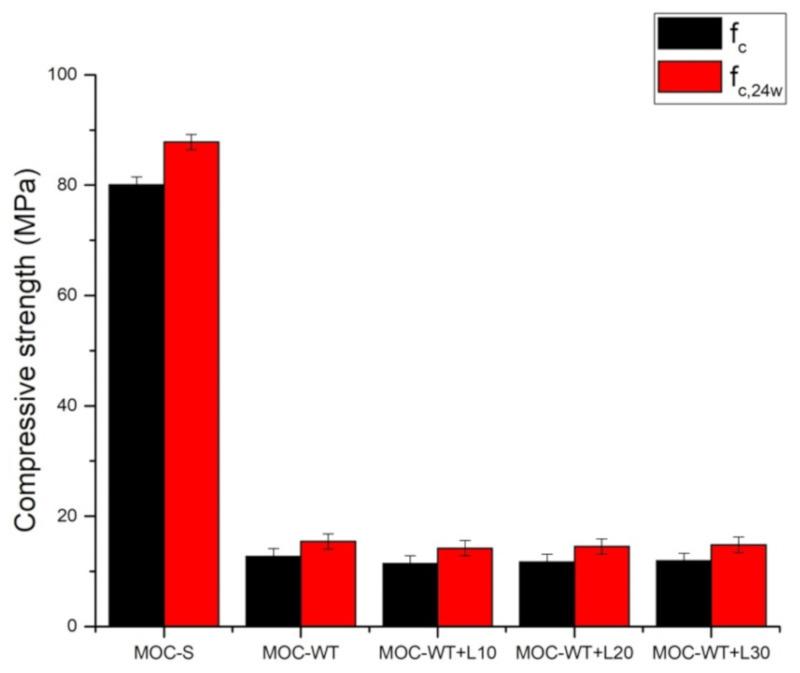
Comparison of the control compressive strength and that of samples exposed to water.

**Figure 7 materials-13-04828-f007:**
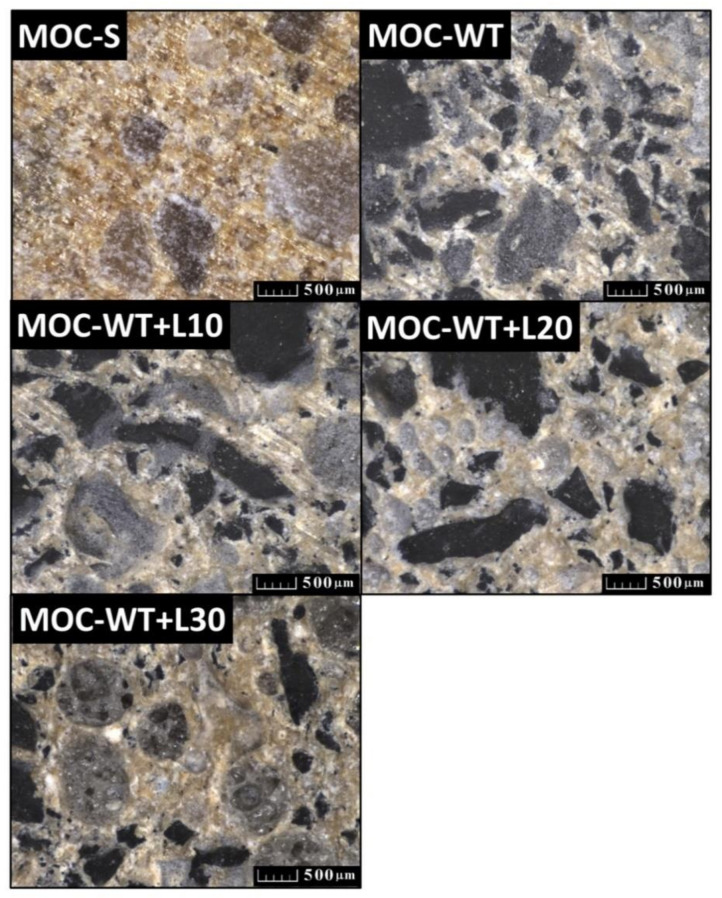
Photographs of composites acquired by optical microscopy (scale bar: 500 μm).

**Table 1 materials-13-04828-t001:** The weight proportioning of designed composites (g), the dosage of MgO, MgCl_2_∙6H_2_O, and water was kept constant for all produced mixtures, and was 1350 g, 597.9 g, and 496.8, respectively.

Composite	Silica Sand	Rubber Granulate	Liaver
MOC-S	3 × 1080	-	-
MOC-WT	-	3 × 350	-
MOC-WT+L10	-	3 × 315	72.4
MOC-WT+L20	-	3 × 279.9	144.9
MOC-WT+L30	-	3 × 244.9	217.3

**Table 2 materials-13-04828-t002:** The specific density of applied aggregates.

Filler	Specific Density *ρ*_sf_ (kg∙m^−3^)
Silica Sand	2652
Granulated rubber	
0–1 mm	1195
1–2 mm	1172
Liaver	
0.1–0.3 mm	1240
0.25–0.5 mm	843
0.5–1.0 mm	717
1.0–2.0 mm	579

**Table 3 materials-13-04828-t003:** Powder density and thermal properties of aggregates.

Aggregate	State	Powder Density *ρ*_p_ (kg∙m^−3^)	Thermal Conductivity *λ*_f_ (W∙m^−1^∙K^−1^)	Volumetric Heat Capacity *C*_wf_ × 10^6^ (J m^−3^∙K^−1^)
Silica Sand	loose	1363	0.399	0.256
	compacted	1910	0.539	0.324
Granulated rubber	loose	441	0.085	0.270
	compacted	517	0.096	0.290
Liaver	loose	305	0.080	0.237
	compacted	346	0.094	0.249

**Table 4 materials-13-04828-t004:** Workability of fresh mixtures and macrostructural parameters of hardened composites.

Materials	Spread Diameter (mm)	*ρ*_s_ (kg∙m^−3^)	*ρ*_b_ (kg∙m^−3^)	*Ψ* (%)
MOC-S	160/165 ± 5	2320 ± 28	2165 ± 30	6.7 ± 0.1
MOC-WT	125/125 ± 5	1671 ± 20	1440 ± 20	13.8 ± 0.3
MOC-WT+L10	130/130± 5	1662 ± 20	1413 ± 20	15.0 ± 0.3
MOC-WT+L20	130/135 ± 5	1654 ± 20	1365 ± 19	17.5 ± 0.4
MOC-WT+L30	135/135 ± 5	1609 ± 19	1316 ± 18	18.2 ± 0.4

**Table 5 materials-13-04828-t005:** Mechanical parameters of hardened composites.

Materials	*f*_f_ (MPa)	*f*_c_ (MPa)	*E_d_* (GPa)
MOC-S	18.9 ± 0.3	80.1 ± 1.1	44.5
MOC-WT	4.5 ± 0.1	12.7 ± 0.2	8.6
MOC-WT+L10	3.8 ± 0.1	11.4 ± 0.2	9.4
MOC-WT+L20	4.1 ± 0.1	11.7 ± 0.2	9.6
MOC-WT+L30	5.3 ± 0.1	11.9 ± 0.2	10.0

**Table 6 materials-13-04828-t006:** Hygric parameters of hardened composites.

Materials	*W*_a_ (%)	*A*_w_ ×10^−4^ (kg∙m^−2^s^−1/2^)	*w*_cap_ (kg∙m^−3^)	*D*_app_ ×10^−9^ (m^2^∙s^−1^)
MOC-S	1.33 ± 0.02	6.0 ± 0.1	50.3 ± 0.9	0.14 ± 0.03
MOC-WT	2.01 ± 0.02	13.3 ± 0.3	63.5 ± 1.1	0.44 ± 0.10
MOC-WT+L10	2.45 ± 0.03	13.7 ± 0.3	74.7 ± 1.3	0.33 ± 0.08
MOC-WT+L20	2.51 ± 0.03	14.0 ± 0.3	75.5 ± 1.4	0.34 ± 0.08
MOC-WT+L30	2.86 ± 0.03	19.6 ± 0.5	79.7 ± 1.4	0.61 ± 0.14

**Table 7 materials-13-04828-t007:** Thermophysical properties of hardened composites.

Materials	*λ* (W∙m^−1^∙K^−1^)	*a* ×10^−6^ (m^2^∙s^−1^)	*C*_v_ ×10^6^ (J∙m^−3^∙K^−1^)
MOC-S	3.247	1.455	2.233
MOC-WT	0.787	0.384	2.055
MOC-WT+L10	0.771	0.391	1.977
MOC-WT+L20	0.751	0.432	1.743
MOC-WT+L30	0.731	0.426	1.716

**Table 8 materials-13-04828-t008:** The water resistance coefficient.

Materials	*α*_w_ (GPa)
MOC-S	1.1
MOC-WT	1.2
MOC-WT+L10	1.3
MOC-WT+L20	1.2
MOC-WT+L30	1.2

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
