# Peer review of "Magnesium Oxychloride Cement Composites Lightened with Granulated Scrap Tires and Expanded Glass"

_materials, 2020, doi:10.3390/ma13214828_

Round 1

Reviewer 1 Report

The Authors report a study about the application of granulated scrap tires in the formulation of composites alternative to the known Portland cements.

The manuscript is well written and the results clearly presented and discussed. The data reported are of interest, especially for industrial application.

Two minor suggestions:

- Table 1. I would remove the first three columns as the amounts of MgO, MgCl26H2O, and water are the same for all the samples. I would add a note to the table caption.

- Figure 4. Please homogenize the name of the curves: in the inbox the three curves are named “silica sand”, “waste tire rubber”, and  “Liaver”; while under the table the names are “silica sand”, “rubber”, and “expanded glass granulates”.

Author Response

The Authors report a study about the application of granulated scrap tires in the formulation of composites alternative to the known Portland cements.

The manuscript is well written and the results clearly presented and discussed. The data reported are of interest, especially for industrial application.

Thank you for such positive review of our paper. Your work on review of our manuscript is highly appreciated.  We considered both your suggestions and corrected the paper as recommended. We performed also final reading, checked references, cleaned typos we found and grammatically corrected the revised manuscript. The revised text is in the manuscript marked up by a yellow colour and the particular changes are described in comments bellow.  

Two minor suggestions:

- Table 1. I would remove the first three columns as the amounts of MgO, MgCl26H2O, and water are the same for all the samples. I would add a note to the table caption.

The text and Table 1 were revised as recommended.

- Figure 4. Please homogenize the name of the curves: in the inbox the three curves are named “silica sand”, “waste tire rubber”, and  “Liaver”; while under the table the names are “silica sand”, “rubber”, and “expanded glass granulates”.

As suggested, the names of the curves were uniformed.

Reviewer 2 Report

The article contains the results of experimental investigation of novel lighweight composites for construction. Below I present some minor remarks:

1. Please, add the reference about wave velocity propagation and elastic modulus of the investigated samples (Equation 2).

2. The results of the ultrasonic tests usually differ significantly form the results of the destructive tests. The concrete mesostructure has an impact on wave propagation phenomenon (please, check the recent literature). Could you comment why the results of destructive tests are not included and discussed in the article?

Author Response

The article contains the results of experimental investigation of novel lightweight composites for construction. Below I present some minor remarks:

Thank you for such positive review of our paper. Your work on review of our manuscript is highly appreciated.  We considered both your suggestions and corrected the paper as recommended. We performed also final reading, checked references, cleaned typos we found and grammatically corrected the revised manuscript. The revised text is in the manuscript marked up by a yellow colour and the particular changes are described in comments bellow. 

1. Please, add the reference about wave velocity propagation and elastic modulus of the investigated samples (Equation 2).

The reference no. [39] was newly added. The text and list of references was revised accordingly.

2. The results of the ultrasonic tests usually differ significantly form the results of the destructive tests. The concrete mesostructure has an impact on wave propagation phenomenon (please, check the recent literature). Could you comment why the results of destructive tests are not included and discussed in the article?

Thank you for this comment. You are right. The static and dynamic elastic modulus significantly differs. We checked the literature, but the data on the comparison of the static elastic modulus and dynamic elastic modulus of MOC-based materials are not available (several papers report on this comparison for concrete, rocks, etc.- reduction coefficient can be used for recalculation of dynamic elastic modulus to static elastic modulus). We have chosen the dynamic measurement, as it is fast and non-destructive method and our laboratory is not equipped with the apparatus for the strain assessment in the static elastic modulus test. We believe the ultrasound technique is very useful as it has become popularly used among other fields in composite materials, biomaterials, building materials, etc. In anyway, we will improve our measurement unit and present both static and dynamic elastic moduli in our future research.

Reviewer 3 Report

Line 42:  ETRMA : What is the long form?

Line 72: In the last paragraph of the introduction the authors cite that they aim the possible use of granulated waste tire rubber in composition of lightweight composites made of magnesium oxychloride cement. However, “the possible use” expression is too vague too broad. They must be more specific.

Line 108: If you have a laser particulate size measurement system, it would be useful to show the size distribution of the recycled rubbers.

Figure 2, scale should be visible and legible.

Line 129 : loos? bulk density 

Table 1 is hard to follow. Is there any specific reason not to use mass fractions instead of specific mass?

Line 169 “flexural strength test was determined >> "test" is unnecessary and “banding” should be “bending”

Line 173 dynamic modulus of elasticity is not the same as elastic modulus

Line 178 “apparent from” : not correct use of English

Figure 6, the images in the figure are given randomly. They should be well specified and cited.

This manuscript treats an interesting topic which can contribute to the alternative building materials field. It is generally well written. However, the structure of the paper is weak and it is hard to follow the original side of the study due to many not well explained details. The story which lies behind the article is not well presented. Many experimental studies were performed but their necessity did not well explain. Moreover, the manuscript did not focus on a specific point, instead it remains too broad.

Even if the manuscript treats an interesting subject with experimental evidence, due to the lack of structure and the fact that it does not address a specific problem, I would not recommend acceptance of this manuscript for publication.

Author Response

Line 42:  ETRMA : What is the long form?

Abbreviation ETRMA stands for the European Tyre & Rubber Manufactures’ Association (see reference [3]). This information was provided and the manuscript revised. Thank you for this suggestion.

Line 72: In the last paragraph of the introduction the authors cite that they aim the possible use of granulated waste tire rubber in composition of lightweight composites made of magnesium oxychloride cement. However, “the possible use” expression is too vague too broad. They must be more specific.

The text was revised accordingly and the reasons of the use of waste tire granulate in composition of the examined composites were given in the text below. The main benefits of the rubber aggregate were: low weight, i.e. improvement of thermal performance, hydropobic nature of rubber particles – increase in water resistance, reuse of waste in production of novel products (ecological and economic benefits).

Line 108: If you have a laser particulate size measurement system, it would be useful to show the size distribution of the recycled rubbers.

Our laboratory is equipped with the Laser Particle Sizer ANALYSETTE 22 MicroTec Plus with the measuring range 0.08 – 2000 μm. However, as the rubber, sand, and Liaver had particles in the range 0-2 μm, we have rather used standard sieve analysis as the bigger particles were in the boundary of the detectable range. Typically, for particles < 1 μm we apply laser diffraction, but it was not this case.  

Figure 2, scale should be visible and legible.

The scale of Fig. 2 was corrected as recommended.

Line 129 : loos? bulk density

The text was revised and term “loose“ was removed. For explanation, in some papers the term “loose bulk density“ is used instead of powder density.

Table 1 is hard to follow. Is there any specific reason not to use mass fractions instead of specific mass?

We are used to present specific mass in the description of the composition of studied materials as it was originally used in preparation of samples for the particular tests. The Table 1 was partially modified as suggested Reviewer 1 who reported on our manuscript very positively.

Line 169 “flexural strength test was determined >> "test" is unnecessary and “banding” should be “bending”4

Thank you very much for pointing out this mistake and typo. The text was revised.

Line 173 dynamic modulus of elasticity is not the same as elastic modulus

Thank you for this comment. The text was revised and uniformed. The term dynamic elastic modulus is used throughout the manuscript.

Line 178 “apparent from” : not correct use of English

Figure 6, the images in the figure are given randomly. They should be well specified and cited.

You are right. Organization of the images in Figure 6 was chaotic. The images were reorganized and specified.

This manuscript treats an interesting topic which can contribute to the alternative building materials field. It is generally well written. However, the structure of the paper is weak and it is hard to follow the original side of the study due to many not well explained details. The story which lies behind the article is not well presented. Many experimental studies were performed but their necessity did not well explain. Moreover, the manuscript did not focus on a specific point, instead it remains too broad.

Even if the manuscript treats an interesting subject with experimental evidence, due to the lack of structure and the fact that it does not address a specific problem, I would not recommend acceptance of this manuscript for publication.

Thank you for positive comment on the interest of the paper and its general evaluation. However, similarly as other 2 reviewers whose assessments were positive, we don’t see any lack of the structure of the paper.   The motivation of the paper is well described and quit clear: design and development of lightweight thermal insulation composites with limited water absorption and increased resistance against moisture damage, which was achieved by the use of processed waste tires. The impact of the use of rubber granulated on the properties and performance of the researched materials was studied and the findings were discussed and then summarized in conclusions. We believe, after consideration of all your comments and suggestions and also by other improvements made based on other two reviewers’ reports the revised manuscript is of high quality and interesting for materials research community. Therefore, we consider it worthy to be published in Materials.  

The revised text is in the manuscript marked up by a yellow colour and the particular changes are described in comments above.  We performed also final reading, checked references, cleaned typos we found and grammatically corrected the revised manuscript.

Round 2

Reviewer 3 Report

The authors thoroughly corrected/modified a part of the previously mentioned recommendations which helped to the improvement of the manuscript. However, I stand still behind what I have recommended about the structure and the motivation and I do not see an improvement on that topic. 

Author Response

Thank you for your help in the improvement of our paper and presentation of research conducted. However, similarly as other 2 reviewers whose assessments were positive, we don’t see any lack of the structure of the paper.   We used standard structure of the paper typical for presentation of experimental research – Introduction – state of the art, motivation and novelty of the paper, Materials and methods – design and description of experiments and materials, Results and discussion – presentation of obtained data, its analysis, discussion, and comparison with published papers (if available), Conclusions – summary of results and their highlighting.

As requested, we partially improved the motivation of the paper which is now well described and quit clear: design and development of lightweight thermal insulation composites with limited water absorption and increased resistance against moisture damage, which was achieved by the use of processed waste tires. The impact of the use of rubber granulated on the properties and performance of the researched materials was studied and the findings were discussed and then summarized in conclusions.

We also give short description, why the applied testing methods were used.

The conclusions were slightly modified.

The novelty of the paper was more elaborated.

We believe, after consideration of all your comments and suggestions and also by other improvements made based on other two reviewers’ reports the revised manuscript is of high quality and interesting for materials research community. Therefore, we consider it worthy to be published in Materials. 

The revised text is in the manuscript marked up by a yellow colour (the first revision is not highlighted) and the particular changes are described in comments above.  We performed also final reading, checked references, cleaned typos we found and grammatically corrected the revised manuscript.